# MEGADance: Mixture-of-Experts Architecture for Genre-Aware 3D Dance Generation

**Kaixing Yang**[1]* **Xulong Tang**[3]* **Ziqiao Peng**[1]* **Yuxuan Hu**[1] **Jun He**[1]† **Hongyan Liu**[2]†

[1]Renmin University of China [2]Tsinghua University [3]Malou Tech Inc

## Abstract

Music-driven 3D dance generation has attracted increasing attention in recent years, with promising applications in choreography, virtual reality, and creative content creation. Previous research has generated promising realistic dance movement from audio signals. However, traditional methods underutilize genre conditioning, often treating it as auxiliary modifiers rather than core semantic drivers. This oversight compromises music-motion synchronization and disrupts dance genre continuity, particularly during complex rhythmic transitions, thereby leading to visually unsatisfactory effects. To address the challenge, we propose MEGADance, a novel architecture for music-driven 3D dance generation. By decoupling choreographic consistency into dance generality and genre specificity, MEGADance demonstrates significant dance quality and strong genre controllability. It consists of two stages: (1) High-Fidelity Dance Quantization Stage (HFDQ), which encodes dance motions into a latent representation by Finite Scalar Quantization (FSQ) and reconstructs them with kinematic-dynamic constraints, and (2) Genre-Aware Dance Generation Stage (GADG), which maps music into the latent representation by synergistic utilization of Mixture-of-Experts (MoE) mechanism with Mamba-Transformer hybrid backbone. Extensive experiments on the FineDance and AIST++ dataset demonstrate the state-of-the-art performance of MEGADance both qualitatively and quantitatively. Code is available at https://github.com/XulongT/MEGADance.

## 1 Introduction

Music-to-dance generation is a crucial task that translates auditory input into dynamic motion, with significant applications in virtual reality, choreography, and digital entertainment[1, 2]. By automating this process, it enables deeper exploration of the intrinsic relationship between music and movement[3], while expanding possibilities for creative content generation. Due to its broad impact, music-to-dance generation has attracted increasing attention[4, 5, 6].

Current music-to-dance generation approaches have witnessed rapid progress and can be broadly categorized into two paradigms[4, 7]: (1) One-stage methods directly map musical features to human motion[8, 6, 9]. (2) Two-stage methods first construct choreographic units and then learn their probability distributions conditioned on music [4, 5, 10]. However, previous methods only treat genre as a weak auxiliary bias rather than the core semantic driver[11, 2, 8], facing several essential problems such as misaligned music-motion synchronization and disrupted dance genre continuity. For example, A Uyghur movements clip is inappropriately mixed into the Popping routine, when a typical Popping music exhibits complex transitions in rhythm and intensity, as shown in Fig. 1. This oversight leads to unsatisfactory visual effects, and fails to meet genre-specific user demands.

---

*Equal contribution.
†Corresponding author.

39th Conference on Neural Information Processing Systems (NeurIPS 2025).

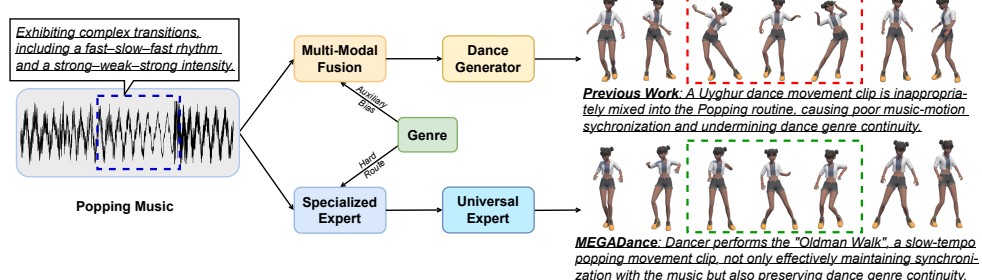

Figure 1: MEGADance enhances choreography consistency by decoupling it into dance generality and genre specificity via the Mixture-of-Experts design. Compared to previous methods, it produces synchronized dance with genre continuity, even under complex music conditions.

To address these limitations, we propose MEGADance, the first Mixture-of-Experts[12] (MoE) architecture for Genre-Aware 3D Dance Generation. By decoupling choreographic consistency into dance generality, modeled by Universal Experts shared across all genres, and genre specificity, captured by Specialized Experts selected via genre-guided hard routing, MEGADance enables robust musical alignment and fine-grained stylistic fidelity. Due to its structured inductive bias, MEGADance exhibits strong genre control and remains robust even in the presence of modality conflicts, such as generating a Breaking dance for soft-paced Chinese music while preserving rhythmic and dynamic alignment.

Specifically, MEGADance comprises two stages. (1) High-Fidelity Dance Quantization Stage (HFDQ), which encodes dance motions into a latent representation. In the HFDQ stage, we introduce Finite Scalar Quantization[13] (FSQ), which replaces traditional VQ-VAE codebooks to mitigate codebook collapse and enhance latent diversity. Additionally, we impose kinematic constraints by simultaneously reconstructing 3D joints through Forward Kinematics[14] from SMPL[14], and dynamic constraints by reconstructing dances while considering velocity and acceleration, to enhance spatio-temporal coherence. (2) Genre-Aware Dance Generation Stage (GADG), which maps music into the latent representation. In the GADG, the Universal Experts and Specialized Experts work synergistically to jointly modeling dance generality and genre specificity. For genre-disentangled expert design, Universal Experts model universal rhythmic and temporal structures across all genres, enhancing robustness to diverse musical inputs and stabilizing cross-modal alignment; and Specialized Experts specialize in fine-grained stylistic variations unique to each genre, guided by hard routing to effectively disentangle genre-dependent features from genre-invariant dynamics. For expert structure, each expert, whether Specialized or Universal, adopts an autoregressive Mamba-Transformer hybrid backbone, combining Mamba's efficient intra-modal local dependency capture [15] with the Transformer's cross-modal global contextual understanding[16], thereby enabling the generation of temporally coherent and musically aligned dance motions.

The contributions of our work can be summarized as: (1) We introduce MEGADance, the first Mixture-of-Experts (MoE) architecture for music-to-dance generation, designed to enhance choreographic consistency by decoupling it into dance generality and genre specificity. MEGADance achieves state-of-the-art (SOTA) performance and demonstrates robust genre controllability, as demonstrated through extensive qualitative and quantitative experiments on the AIST++[1] and FineDance[2] datasets. (2) We propose a High-Fidelity Dance Quantization framework that introduces FSQ with kinematic-dynamic dual constraints, ensuring complete codebook utilization (100% vs. VQ-VAE[17]'s 75%) while achieving excellent reconstruction accuracy. (3) We design a Mamba-Transformer hybrid backbone for music-to-dance generation, combining Mamba's efficient intra-modal local dependency capture with the Transformer's cross-modal global contextual understanding.

## 2 Related Work

### 2.1 One-Stage Music-to-Dance Generation

Music and dance are deeply interconnected, leading to significant advancements in the field of music-driven 3D dance generation. Researchers utilize musical features extracted via tools like Librosa[1],

Jukebox[18], and MERT[19] to predict human motion, including SMPL[14] parameters[1] and body keypoints[4]. Early methods primarily employ encoder-decoder architectures to directly obtain entire human motion sequence [20, 21, 22, 23, 1]. Recognizing the natural hierarchical structure of human joints, some researchers introduced Graph Convolutional Networks (GCNs)[24, 25] to enhance interaction at the joint level, thereby improving the biomedical plausibility of the generated motions. In AIGC, Generative Adversarial Networks (GANs) are widely applied, some researchers introduced it in music-to-dance tasks[26, 9, 27]. Specifically, GANs' generators produce dance motions from music, with discriminators providing feedback to guide generated motions more natural. Recently, Diffusion Models have shown remarkable success in various AIGC tasks, with notable applications extending to the music-to-dance domain[18, 2, 8, 6, 28], but the computational cost of the sampling process remains high, especially in long-sequence generation scenarios for the music-to-dance task. However, the lack of explicit constraints to maintain the generated pose within proper spatial boundaries often leads to nonstandard poses that extend beyond the dancing subspace during inference, resulting in low dance quality in practical applications.

## 2.2 Two-Stage Music-to-Dance Generation

Leveraging the inherent periodicity of dance kinematics, researchers propose Two-Stage methods, including (1) Dance Quantization stage: curating choreographic units from motion databases, and (2) Dance Generation stage: learning music-conditioned probability distributions over these units. As these choreographic units are derived from real human motion data, two-stage approaches naturally benefit from a biomechanical plausibility prior, contributing to the realism of generated dances.

**Dance Quantization Stage.** Traditional methods [29, 30, 31, 26] construct choreographic units through uniform segmentation of motion sequence, incurring high computational overhead. Recent works [32, 33] employ VQ-VAE for intelligent unit construction, significantly reducing time/space complexity. Considering the relative independence of upper and lower body movements, [4, 5] construct choreographic unit for lower and upper parts separately, improving motion reconstruction through expanded unit capacity ($L \rightarrow L \times L$). However, above works predominantly operate on 3D human body keypoints, which lack expressiveness in capturing nuanced motion details. [5, 6] construct choreographic units in the SMPL pose space but apply uniform treatment across joints, neglecting the body's kinematic hierarchy, such as root errors propagate globally through kinematic chains while hand errors remain localized.

**Dance Generation Stage.** To model choreographic unit distributions, Choreomaster [26] employs a GRU-based backbone, while DanceRevolution [9] uses RNNs. Recent works like Bailando [4] and Bailando++[5] adopt cross-modal Transformers for improved temporal modeling and music-motion alignment. Moreover, Everything2Motion [34] and TM2D [32] leverage pretrained models in text-to-motion[33, 35] generation to improve motion quality, often at the expense of choreographic complexity and creativity. To enrich input representations, [2, 8, 31, 7] introduce genre information via shallow fusion, such as cross attention [31] or feature addition [7]. However, these approaches remain insufficient for achieving robust genre controllability, particularly under cross-modal conflicts, such as generating Breaking dances conditioned on slow-tempo traditional Chinese music.

In conclusion, existing methods face two key limitations: (1) VQ-VAE-based quantization suffers from low codebook utilization; (2) Insufficient utilization of genre information results in poor music-motion synchronization and disrupted dance genre continuity. Thus, we propose MEGADance, a two-stage framework. The Dance Quantization stage employs FSQ with kinematic and dynamic constraints to enhance codebook efficiency while preserving reconstruction fidelity. The Dance Generation stage adopts an MoE architecture with a Mamba-Transformer backbone to jointly capture dance generality and genre specificity with efficiency.

# 3 Methodology

## 3.1 Problem Definition

Given a music sequence $M = \{m_0, m_1, ..., m_T\}$ and a dance genre label $g$, our objective is to synthesize the corresponding dance sequence $S = \{s_0, s_1, ..., s_T\}$, where $m_t$ and $s_t$ denotes the music and dance feature at time step $t$. We define each music feature $m_t$ as a 35-dim vector[8] extracted by Librosa[36], including 20-dim MFCC, 12-dim Chroma, 1-dim Peak, 1-dim Beat and

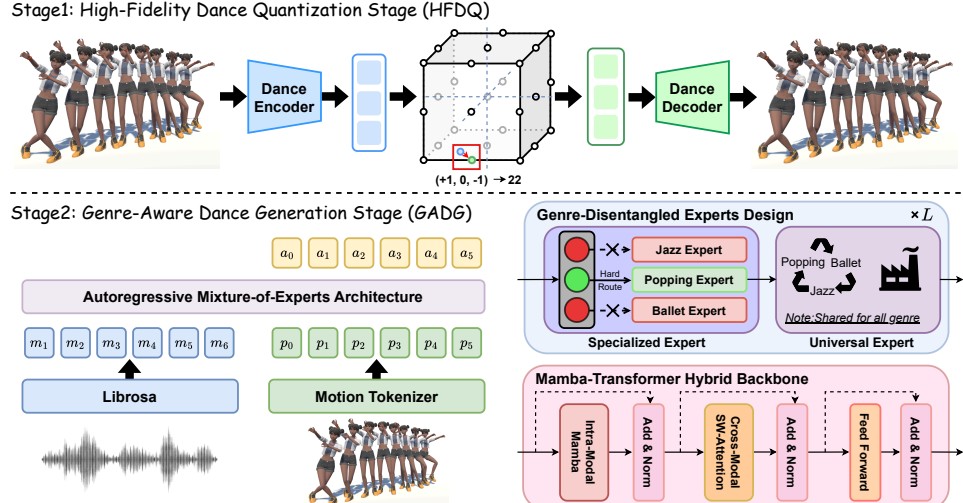

Figure 2: Overview of MEGADance. Stage 1 (HFDQ) quantizes dance into upper/lower-body latent codes using FSQs with kinematic and dynamic reconstruction constraints. Stage 2 (GADG) maps music to codes via an L-layer Mixture-of-Experts architecture (blue: Specialized Experts, pink: Universal Experts), where each expert is equipped with a Mamba-Transformer hybrid backbone

1-dim Envelope. We encode genre label $g$ as a one-hot vector. We represent each dance feature as a 147-dim vector $s_t = [\tau; \theta]$, where $\tau$ and $\theta$ encapsulate the root translation and 6-dim rotation representation[37] of the SMPL[14] model, respectively. Furthermore, we synchronize the music sequence with the dance sequence at a temporal granularity of 30 frames per second.

## 3.2 High-Fidelity Dance Quantization

**Finite Scalar Quantization with Motion Decomposition.** Choreographic units serve as the fundamental building blocks of dance composition, forming the basis for structuring and connecting movements. Despite variations in style and tempo, dances across genres exhibit common underlying units. Our objective is to unsupervisedly encapsulate these units into a versatile and reusable codebook, enabling any dance sequence to be represented as a sequence of discrete codebook elements. To account for the relative independence between upper-body and lower-body movements during dance, we maintain separate codebooks for the upper and lower $\mathcal{Z} = \{\mathcal{Z}_k^u, \mathcal{Z}_k^l\}$ body, where $k$ represent codebook size. Additionally, root translation velocities are associated with the lower body to preserve natural motion dynamics. This decomposition allows the combination of different code pairs to cover a wider array of choreographic units.

Our 3D motion reconstruction approach, illustrated in Fig. 2, initiates with a Dance Encoder $\mathbf{E}$ (a three-layer 1D-CNN for information aggregation and a two-layer MLP for dimension adjustment) encoding the dance sequence $S = \{S^u, S^l\}$ into context-aware features $\mathbf{z} = \{\mathbf{z}^\mathbf{u}, \mathbf{z}^\mathbf{l}\}$. These features are quantized using Finite Scalar Quantization (FSQ) to obtain $\hat{\mathbf{z}} = \{\hat{\mathbf{z}^\mathbf{u}}, \hat{\mathbf{z}^\mathbf{l}}\}$, which are then decoded by Dance Decoder $\mathbf{D}$ (a two-layer MLP for dimension adjustment and a three-layer 1D TransConv for information restoration) to reconstruct the dance movement $\hat{S} = \{\hat{S^u}, \hat{S^l}\}$. To resolve the codebook collapse problem caused by the conventional VQ-VAE[38] based quantization, we adopt FSQ. By replacing the discrete "argmin" codebook selection with scalar quantization via differentiable bounded rounding, FSQ enables balanced utilization and stable gradient propagation:

$$\hat{\mathbf{z}} = f(\mathbf{z}) + \text{sg}\left[\text{Round}[f(\mathbf{z})] - f(\mathbf{z})\right], \tag{1}$$

where $f(\cdot)$ is the bounding function, setting as the $\text{sigmoid}(\cdot)$ function in our practice. Each channel in $\hat{\mathbf{z}}$ will be quantized into one of the unique $L$ integers, therefore we have $\hat{\mathbf{z}} \in \{1, \ldots, L\}^d$. The codebook size is calculated as $k = \prod_{i=1}^{d} L_i$, and $L, d$ are super parameter. In conclusion, FSQ with Motion Decomposition expands its effective motion representation capacity, thereby enhancing the diversity of subsequent generated dance.

**Motion Reconstruction with Kinematic-Dynamic Constraint.** Unlike VQ-VAE requiring additional loss to update any extra lookup codebook, FSQ directly integrates numerical approximations "round" within its workflow. The Dance encoder $E$ and decoder $D$ are simultaneously learned with the codebook via the following loss function:

$$\mathcal{L}_{FSQ} = \mathcal{L}_{\text{smpl}}(\hat{S}, S) + \mathcal{L}_{\text{joint}}(\hat{J}, J). \tag{2}$$

The $\mathcal{L}_{\text{smpl}}$ is the reconstruction loss that ensures the predicted 3D SMPL sequence closely aligns with the ground truth. Simple reconstruction on SMPL parameters treats all joints equally, neglecting the complex hierarchical tree structure of human body joints, different joints vary in their tolerance to errors. For instance, errors at the root node propagate throughout all nodes, whereas errors at the hand node primarily affect only itself. Thus, we execute Forward Kinematic[14] techniques to derive 3D joints and apply reconstruction constraints $\mathcal{L}_{\text{joint}}$ between $\hat{J}$ and $J$. Moreover, the reconstruction loss accounts not only for the spatial positions but also for the velocities ($\alpha_1$) and accelerations ($\alpha_2$) of the movements, where $'$ and $''$ correspond to the first- and second-order derivatives respectively:

$$
\begin{aligned}
\mathcal{L}_{\text{smpl}}(\hat{S}, S) &= \|\hat{S} - S\|_1 + \alpha_1\|\hat{S}' - S'\|_1 + \alpha_2\|\hat{S}'' - S''\|_1, \\
\mathcal{L}_{\text{joint}}(\hat{J}, J) &= \|\hat{J} - J\|_1 + \alpha_1\|\hat{J}' - J'\|_1 + \alpha_2\|\hat{J}'' - J''\|_1.
\end{aligned}
\tag{3}
$$

Through training, our method facilitates the interchangeability of orthographic memory codes, enabling the synthesis of new motions from existing choreographic units by recombining different code elements.

### 3.3 Genre-Aware Dance Generation

With dance sequences represented as discrete latent codes, the music-to-dance generation task is simplified from a regression problem into a classification problem, where the goal is to select appropriate pose codes from a codebook rather than predict continuous motion parameters.

#### 3.3.1 Mixture-of-Experts Architecture

As illustrated in Fig. 2, we perform cross-modal autoregressive generation. Given music features $m_{1:T}$ extracted using Librosa, dance genre label $g$, and the previous pose codes $p_{0:T-1} = \{p_{0:T-1}^l, p_{0:T-1}^u\}$ encoded by the Motion Tokenization $\mathbf{E}$ in HFDQ, the GADG predicts action probabilities $a_{0:T-1} = \{a_{0:T-1}^l, a_{0:T-1}^u\}$ of every $z_i \in \mathcal{Z}$, using an $L$-layer MoE architecture. To align the predicted action probabilities $a_{0:T-1}$ with the next pose codes $p_{1:T}$, we employ a supervised Cross-Entropy[4] loss, where each predicted action $a_t$ is matched to its corresponding target pose code $p_{t+1}$. The inference of GADG includes: 1) Short sequences ($\leq 5.5s$) via autoregressive generation, 2) Long sequences via sliding-window prediction with 5.5s overlap.

Specifically, each MoE layer contains a Specialized Expert and a Universal Expert, which jointly model dance generality and genre specificity. Specialized Experts (e.g., Pop Expert, Jazz Expert) are conditionally activated based on the genre label $g$, and input features are routed to the corresponding expert via a hard routing mechanism. In parallel, features from all genres are processed by the shared Universal Expert to capture genre-invariant dynamics. For expert structure, each expert, whether Specialized or Universal, adopts an autoregressive Mamba-Transformer hybrid backbone: Mamba captures intra-modal local dependencies, while Transformer encodes cross-modal global context, thereby enabling the generation of temporally coherent and musically aligned dance motions.

#### 3.3.2 Genre-Disentangled Experts Design

**Specialized Experts.** The Specialized Experts are designed to capture genre-specific stylistic patterns, motivated by two core considerations: *(1) Structural Inductive Bias:* By isolating parameters across experts, the model enforces separation of genre-specific motion motifs (e.g., Krump's grounded explosiveness vs. Contemporary's fluid transitions), thereby preserving distinct stylistic representations. This separation also introduces genre-aware control priors that mitigate cross-genre interference, which is critical for genre-conditioned dance generation. *(2) Computational Efficiency:* Leveraging sparse MoE design [39], each input is routed to a single expert, significantly reducing parameter redundancy and computational cost.

**Universal Experts.** The Universal Expert learns generalizable representations to complement Specialized Experts through two key roles: *(1) Fundamental Choreographic Prior*: It learns shared low-level patterns across genres (e.g. periodicity, beat synchronization, and biomechanical consistency). In contrast, models relying solely on Specialized Experts often fail under modality mismatch (e.g., producing static or repetitive movements when Ballet music is processed by a Popping Expert). The Universal Expert provides a genre-agnostic prior that enhances stability and expressiveness under complex input conditions. *(2) Disentangled Representation*: By disentangling shared and genre-specific factors, the model allows each expert to specialize in distinct subspaces, enhancing generation quality[40].

### 3.3.3 Mamba-Transformer Hybrid Backbone

**Cross-Modal Global-Context Modeling.** Leveraging the Transformer's global receptive field [16], we concatenate cross-modal features along the temporal axis and facilitate structured interactions among music, upper-body, and lower-body representations, by a Attention layer and a Feed Forward layer. The attention layer is the core component that defines the computational dependencies among sequential data elements and is implemented as:

$$\text{Attention}(Q, K, V, M) = \text{softmax}\left(\frac{QK^T + M}{\sqrt{C}}\right)V, \tag{4}$$

where $Q, K, V$ denote the query, key, and value from input, and $M$ is the mask, which determines the type of attention layers. The two most common attention types are full attention [16], which enables global context exchange, and causal attention [41], which restricts each position to attend only to current and past inputs. Given that music-to-dance generation is typically applied to long sequence while being constrained by limited computational resources, training is commonly conducted on short clips. During inference, the sequence is first extended autoregressively up to the training length (step 1), and then completed using a sliding window approach for the remaining part (step 2). Training driven by standard causal attention only aligns with step 1 and fails to account for the dominant step 2 during inference, thereby limiting generation performance. To better align training with inference, we introduce a sliding-window attention mechanism that mimics the generation process. The attention mask $M \in \mathbb{R}^{3T \times 3T}$ is structured as a $3 \times 3$ block matrix, where each block is a $T \times T$ sliding-window mask, enabling cross-modal global-context attention.

**Intra-Modal Local-Dependency Modeling.** While the Transformer excels at temporal modeling, it is inherently position-invariant and captures sequence order only through positional encodings [16], which limits its deep understanding of local dependencies. In contrast, music-to-dance generation demands strong local continuity between movements. Owing to its inherent sequential inductive bias, Mamba [15] has demonstrated strong performance in modeling fine-grained local dependencies [42, 43]. We therefore apply independent Mamba to the music, upper-body, and lower-body features respectively, to model their intra-modal local dependencies. Specifically, Selective State Space model (Mamba) incorporates a Selection mechanism and a Scan module (S6) [15] to dynamically select salient input segments for efficient sequence modeling. Unlike the traditional S4 model [44] with fixed parameters $A, B, C$, and scalar $\Delta$, Mamba adaptively learns these parameters via fully-connected layers, enhancing generalization capabilities. Mamba employs structured state-space matrices, imposing constraints that improve computational efficiency. For each time step $t$, the input $x_t$, hidden state $h_t$, and output $y_t$ follow:

$$\begin{aligned} h_t &= \bar{A}_t h_{t-1} + \bar{B}_t x_t, \\ y_t &= C_t h_t, \end{aligned} \tag{5}$$

where $\bar{A}_t, \bar{B}_t, C_t$ are dynamically updated parameters. Through discretization with sampling interval $\Delta$, the state transitions become:

$$\begin{aligned} \bar{A} &= \exp(\Delta A), \\ \bar{B} &= (\Delta A)^{-1}(\exp(\Delta A) - I) \cdot \Delta B, \\ h_t &= \bar{A} h_{t-1} + \bar{B} x_t, \end{aligned} \tag{6}$$

where $(\Delta A)^{-1}$ is the inverse of $\Delta A$, and $I$ denotes the identity matrix. The scan module captures temporal dependencies by applying trainable parameters across input segments.

Table 1: Comparison with SOTAs on the FineDance dataset.

| | Quality | | | Creativity | | | Alignment | User Study | | |
|---|---|---|---|---|---|---|---|---|---|---|
| | $FID_k\downarrow$ | $FID_g\downarrow$ | $FID_s\downarrow$ | $DIV_k\uparrow$ | $DIV_g\uparrow$ | $DIV_s\uparrow$ | $BAS\uparrow$ | $DQ\uparrow$ | $DS\uparrow$ | $DC\uparrow$ |
| GT | 0 | 0 | 0 | 10.98 | 7.45 | 6.07 | 0.215 | 4.39 | 4.35 | 4.48 |
| Bailando++[5] | 54.79 | 16.29 | 8.42 | 6.18 | 5.98 | 4.73 | 0.213 | 3.85 | 3.50 | 3.82 |
| FineNet[2] | 65.15 | 23.81 | 13.22 | 5.84 | 5.19 | 4.29 | 0.219 | 3.62 | 3.65 | 3.47 |
| Lodge[8] | 55.03 | 14.87 | 5.22 | 6.14 | 6.18 | 5.50 | 0.218 | 4.18 | 4.17 | 4.08 |
| **MEGADance** | **50.00** | **13.02** | **2.52** | **6.23** | **6.27** | **5.78** | **0.226** | **4.25** | **4.30** | **4.23** |

## 4 Experiment

### 4.1 Dataset

**1) FineDance.** *FineDance* [2] is the largest public dataset for 3D music-to-dance generation, featuring professionally performed dances captured via optical motion capture. It provides 7.7 hours of motion data at 30 fps across 16 distinct dance genres. Following [8], we evaluate on test-set music clips, generating 1024-frame (34.13s) dance sequences. **2) AIST++.** *AIST++* [1] is a widely used benchmark comprising 5.2 hours of 60 fps street dance motion, covering 10 dance genres. Following [1], we use test-set music clips to generate 1200-frame (20.00s) sequences.

### 4.2 Quantitative Evaluation

**Comparison.** We evaluate MEGADance against state-of-the-art (SOTA) baselines on both the FineDance and AIST++ datasets using a comprehensive suite of metrics. For each generated sequence, we compute Fréchet Inception Distance ($FID$) and Diversity ($DIV$) across three feature spaces: (1) Kinetic (k), capturing motion dynamics; (2) Geometric (g), encoding spatial joint relations; and (3) Style (s), extracted via a Transformer-based genre classifier. We also assess music-motion synchronization using Beat Align Score ($BAS$), following [1, 8]. All baseline results are reproduced under our experimental setup to ensure fair comparison. On the FineDance dataset (Tab. 1), MEGADance outperforms all baselines, achieving the lowest FID in all three feature types ($FID_k$=50.00, $FID_g$=13.02, $FID_s$=2.52), the highest motion diversity ($DIV_k$=6.23, $DIV_g$=6.27, $DIV_s$=5.78), and the best BAS (0.226). On the AIST++ datasets (Tab. 2), MEGADance again ranks first in FID ($FID_k$=25.89, $FID_g$=12.62), and achieves strong performance in diversity ($DIV_g$=5.84, $DIV_s$=6.23) and BAS (0.238). These results underscore the effectiveness of our genre-aware Mixture-of-Experts design in balancing motion quality, creativity, and synchronization across diverse datasets. Performance efficiency analysis is provided in Appendix A.3.

**User Study.** Dances inherent subjectivity makes user feedback essential for evaluating generated movements[45] in the music-to-dance generation task. We select 30 in-the-wild music segments (34 seconds each) and generate dance sequence using above models. These sequences are evaluated through a double-blind questionnaire, by 30 participants with backgrounds in dance practice, including undergraduate and graduate-level students. The questionnaires are based on a 5-point scale (Great, Good, Fair, Bad, Terrible) and assess three aspects: Dance Synchronization ($DS$, alignment with rhythm and style), Dance Quality ($DQ$, biomechanical plausibility and aesthetics), and Dance Creativity ($DC$, originality and range). As shown in Tab. 1, MEGADance significantly outperforms all baselines across user-rated metrics (i.e., $DS = 4.30$, $DQ = 4.25$, $DC = 4.23$). Its high scores in various aspects underscore its superiority in generating movements in terms of human preferences.

### 4.3 Qualitative Evaluation

To assess the visual quality of the generated dance sequences, we perform a qualitative comparison between MEGADance and several existing baseline models, as depicted in Fig. 3. In terms of expressiveness, MEGADance outperforms the competing methods in several key areas. For instance, Lodge[8] struggles with stylistic consistency, often blending conflicting dance genres, such as incorporating Uyghur movements into a typical Breaking routine, leading to a disjointed aesthetic. FineNet[2], while capable of generating movement sequences, suffers from significant artifacts, including unnatural sliding and teleportation, which detract from the fluidity and physical realism of the motions. Additionally, Bailando++[5] demonstrates a lack of diversity, with movements fre-

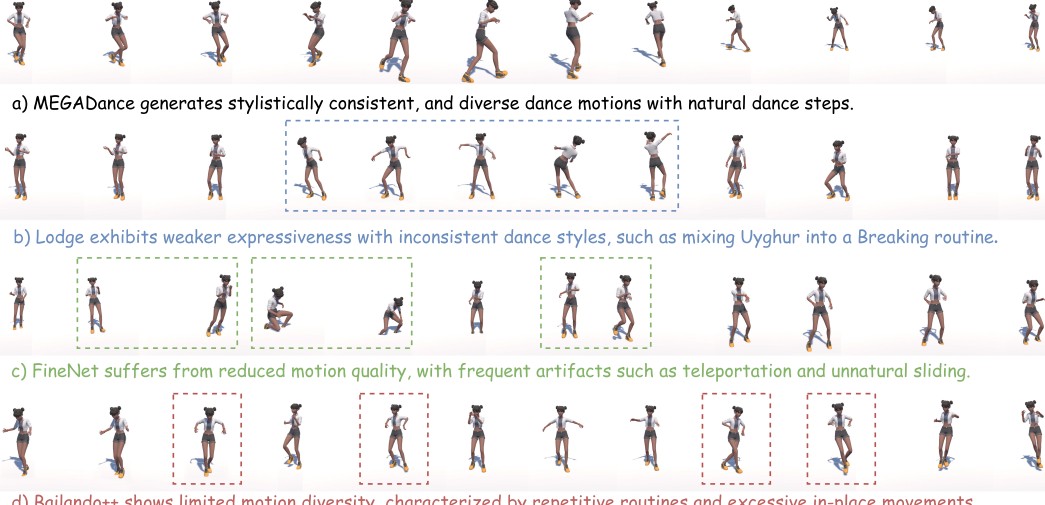

a) MEGADance generates stylistically consistent, and diverse dance motions with natural dance steps.

b) Lodge exhibits weaker expressiveness with inconsistent dance styles, such as mixing Uyghur into a Breaking routine.

c) FineNet suffers from reduced motion quality, with frequent artifacts such as teleportation and unnatural sliding.

d) Bailando++ shows limited motion diversity, characterized by repetitive routines and excessive in-place movements.

Figure 3: Qualitative Analysis on a typical Breaking Battle music clip.

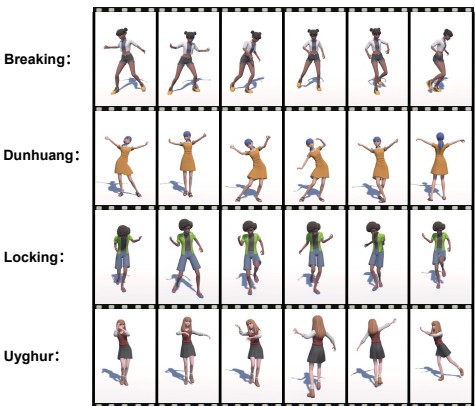

Figure 4: Visualization of Genre Controllability on a representative Chinese music clip.

Table 2: Comparison on the AIST++ Dataset.

|  | $\text{FID}_k\downarrow$ | $\text{FID}_g\downarrow$ | $\text{DIV}_k\uparrow$ | $\text{DIV}_g\uparrow$ | BAS↑ |
|---|---|---|---|---|---|
| GT | 0 | 0 | 9.04 | 7.52 | 0.232 |
| FACT[1] | 35.35 | 22.11 | 5.94 | 6.18 | 0.221 |
| Bailando++[5] | 30.21 | 15.48 | 5.35 | 5.13 | 0.228 |
| EDGE[18] | 42.16 | 22.12 | 3.96 | 4.61 | 0.233 |
| Lodge[8] | 35.72 | 17.92 | 5.72 | 5.91 | **0.247** |
| **MEGADance** | **25.89** | **12.62** | 5.84 | **6.23** | 0.238 |

Table 3: Comparison for Genre Controllability.

|  | $\text{FID}_s\downarrow$ | $\text{DIV}_s\uparrow$ | ACC↑ | F1↑ |
|---|---|---|---|---|
| GT | 0 | 6.07 | 78.31 | 76.35 |
| FineNet[2] | 13.22 | 4.29 | 42.06 | 37.44 |
| Lodge[8] | 5.22 | 5.50 | 51.86 | 45.23 |
| **MEGADance** | **2.52** | **5.78** | **75.64** | **70.81** |

quently repeating and a heavy reliance on static, in-place gestures, limiting the range of expressive movement patterns. These findings underscore the superiority of MEGADance in generating diverse, genre-consistent, and musically synchronized dance sequences.

## 4.4 Genre Controllability Evaluation

To quantitatively assess genre controllability, we compare MEGADance with Lodge [8] and FineNet [2]. Using ground-truth genre labels, we evaluate style alignment ($FID_s$) and style diversity ($DIV_s$) on 20 test clips. We further assess genre classification accuracy ($ACC$) and F1 score ($F1$), conditioned on the correct genre and four randomly sampled negative genres. As shown in Tab. 3, MEGADance achieves the best performance across all metrics. It significantly reduces $FID_s$ (2.52) while improving diversity ($DIV_s$ = 5.78). Despite potential cross-modal conflicts (e.g., assigning a Popping genre to a typical Chinese music clip), MEGADance achieves high genre discriminability ($ACC$ = 75.64, $F1$ = 70.81), closely approaching ground-truth performance. Compared to FineNet and Lodge, our method produces motion sequences that are both more stylistically coherent and genre-distinctive. Our MoE-based genre routing prevents cross-genre interference via disentangled expert subspaces activated by discrete labels, whereas naive continuous fusion (e.g. Feature Addition in [2, 8] or Cross Attention in [31]) inherently blurs stylistic boundaries.

To explore genre controllability from a visual perspective, we assign different dance genres (distal: Breaking/Locking, proximal: Dunhuang/Uyghur) to a representative Chinese music clip. We recommend watching the supplementary video for more details. As shown in Fig. 4, the generated motions

Table 4: Ablation study of the two-stage MEGADance architecture on the FineDance dataset.

| | $FID_k\downarrow$ | $FID_g\downarrow$ | $FID_s\downarrow$ | BAS↑ |
|---|---|---|---|---|
| GT | 0 | 0 | 0 | 0.215 |
| w/o SE | 53.05 | 19.26 | 7.95 | 0.218 |
| w/o UE | 54.50 | 15.52 | 2.91 | 0.223 |
| w/o Mamba | 56.29 | 14.51 | 2.67 | 0.221 |
| **Ours** | **50.00** | **13.02** | **2.52** | **0.226** |

(a) Genre-Aware Dance Generation Stage.

| | SMPL | | Joint | |
|---|---|---|---|---|
| | MSE↓ | MAE↓ | MSE↓ | MAE↓ |
| w/o Kin. Loss | 0.0238 | 0.0847 | 0.0089 | 0.0507 |
| w/o Dyn. Loss | 0.0201 | 0.0779 | 0.0073 | 0.0482 |
| FSQ → VQ-VAE | 0.0308 | 0.0984 | 0.0220 | 0.0842 |
| **Ours** | **0.0200** | **0.0770** | **0.0069** | **0.0469** |

(b) High-Fidelity Dance Quantization Stage.

exhibit both genre fidelity and music synchrony: (1) Breaking: agile footwork with rapid steps and directional shifts, driven by percussive rhythms and dynamic weight transfers; (2) Locking: exaggerated arm swings and torso isolations, punctuated by syncopated, guitar-mimicking gestures; (3) Dunhuang: fluid upper-body arcs with slow rotations and knee undulations, mirroring melodic phrasing and visual symmetry; (4) Uyghur: rapid spins with hand-to-face motifs, emphasizing rotational clarity and rhythmic precision.

### 4.5 Ablation Study

#### 4.5.1 Genre-Aware Dance Generation Stage

We conduct an ablation study to evaluate the contribution of three core components in the Genre-Aware Dance Generation stage: **Specialized Experts (SE)**, **Universal Experts (UE)**, and the **Intra-Modal Local-Dependency Modeling (Mamba)**, with results summarized in Tab. 4a. We recommend watching the supplementary video. **(1) Specialized Experts (SE).** Replacing the SE results in substantial performance degradation across all metrics, especially on $FID_s$ (7.95 vs. 2.52), confirming its critical role in preserving stylistic fidelity across genres. **(2) Universal Expert (UE).** Removing the UE leads to clear drops in $FID_k$ (54.50 50.00) and $FID_g$ (15.52 13.02), while having only minor impact on $FID_s$ and $BAS$. This suggests the UEs effectiveness in providing generalizable priors that enhance structural and dynamic consistency. **(3) Mamba.** Replacing the Mamba in backbone results in moderate performance declines on all metrics (e.g., $FID_k$ from 50.00 to 56.29), demonstrating Mambas advantage in modeling fine-grained local dependencies and improving overall motion quality.

#### 4.5.2 High-Fidelity Dance Quantization Stage

We investigate the effectiveness of three key components in High-Fidelity Dance Quantization Stage: Finite Scalar Quantization (FSQ), the Kinematic Loss ($\mathcal{L}_{Kin.}$), and the Dynamic Loss ($\mathcal{L}_{Dyn.}$). Tab. 4b reports the $MSE$ and $MAE$ on both SMPL parameters and 3D joint positions. **(1) FSQ.** Replacing FSQ with VQ-VAE leads to a significant performance drop in all metrics (e.g., Joint $MSE$ increases from 0.0069 to 0.0220), validating FSQs superiority. Moreover, replacing VQ-VAE with FSQ achieves full codebook utilization (75% 100%). **(2) Kinematic Loss.** Removing the kinematic loss notably increases errors in both SMPL ($MSE$: 0.0200 0.0238) and joint space ($MAE$: 0.0469 0.0507), highlighting its role in enforcing accurate structural constraints via forward kinematics. **(3) Dynamic Loss.** Excluding the dynamic loss results in a moderate degradation in temporal fidelity (e.g., Joint $MSE$: 0.0069 0.0073), demonstrating its contribution for temporal fidelity.

## 5 Conclusion

In this paper, we present MEGADance, a genre-aware MoE-based architecture for music-to-dance generation. MEGADance enhances choreography consistency by decoupling it into dance generality and genre specificity via an MoE design. Through the synergy of high-fidelity dance quantization stage and genre-adaptive dance generation stage, MEGADance achieves state-of-the-art performance and strong genre controllability. In future work, we plan to extend MEGADance with text conditioning to enable more interactive and flexible dance generation.

## 6 Acknowledgements

This work was supported by National Natural Science Foundation of China (NSFC) under Grant Nos. 72572090, 62436010, 62572474 and 62172421.

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

# Appendix

## A.1 Implementation Details

**High-Fidelity Dance Quantization.** In the High-Fidelity Dance Quantization Stage, we use a shared codebook configuration for the upper and lower body branches. The model is trained on 8-second SMPL 6D rotation sequences sampled at 30fps, where $S, \hat{S} \in \mathbb{R}^{T \times 147}$ (i.e., $T = 240$). For data construction, we augment the training set using a sliding window approach with a window size of 240 and a stride of 16. A three-layer CNN encoder $E$ performs temporal downsampling, and a three-layer transposed convolution decoder $D$ performs upsampling. The latent codes for the lower and upper body are $p^l, p^u \in \mathbb{R}^{T'}$, with $T' = 30$. In the Finite Scalar Quantization module, the codebook size is 4375, with $L = [7, 5, 5, 5, 5]$, and the feature dimension is set to 512. For reconstruction, we use both SMPL-parameter loss $\mathcal{L}_{\text{smpl}}$ and joint-position loss $\mathcal{L}_{\text{joint}}$, with velocity and acceleration terms weighted by $\alpha_1 = 0.5$ and $\alpha_2 = 0.25$, respectively. The model is trained for 200 epochs using the Adam optimizer, with exponential decay rates of 0.5 and 0.99 for the first and second moment estimates. A fixed learning rate is used with a batch size of 32. The experimental setup is consistent across FineDance and AIST++.

**Genre-Aware Dance Generation.** In the Genre-Aware Dance Generation Stage, we adopt a Mamba-Transformer hybrid architecture, trained on latent codes $p^l, p^u \in \mathbb{R}^{30}$ extracted from the High-Fidelity Dance Quantization Stage, using 8-second dance sequences at 30fps. For data construction, we augment the training set using a sliding window approach with a window size of 240 and a stride of 16. In MEGADance, the Music Encoder consists of $L = 6$ processing layers. The Mamba block is configured with a model dimension of 512, state size of 16, convolution kernel size of 4, and expansion factor of 2. The Transformer block uses a hidden size of 512, 8 attention heads, a feedforward dimension of 2048, and a dropout rate of 0.25. For Slide Window Attention, we set the autoregressive step to 22 and the sliding window step to 8 to construct the attention matrix. For input representation, genre labels (16 classes from FineDance) are embedded using `nn.Embedding` to match the 512-dimensional latent space, while music features extracted by Librosa (35 dimensions) are projected to 512 dimensions via a two-layer MLP. For output, MEGADance predicts 4375-class distributions via softmax for upper-body and lower-body codebook respectively. The model is optimized using Adam with exponential decay rates of 0.9 and 0.99 for the first and second moment estimates, respectively, trained for 80 epochs with a fixed learning rate and a batch size of 64. The experimental setup is consistent across FineDance and AIST++.

## A.2 Qualitative Analysis for Ablation Study

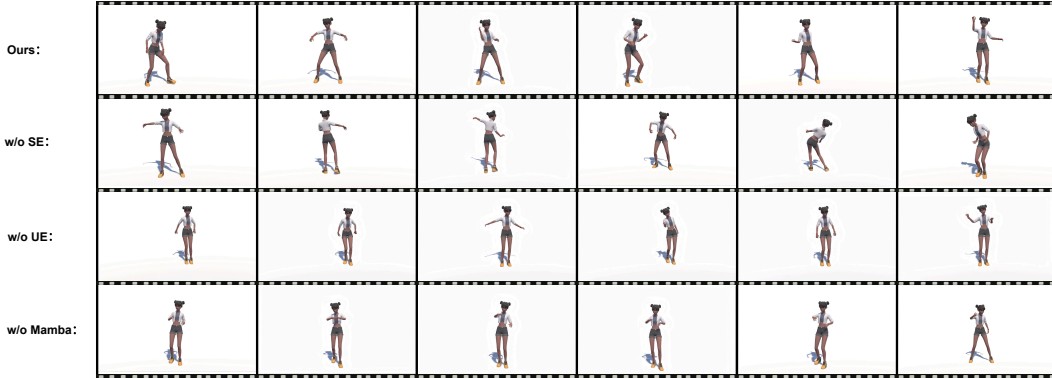

Figure 5: Qualitative Analysis for Ablation Study. MEGADance generates visually expressive dance motions, outperforming others in terms of stylistic consistency and movement diversity.

In this section, we conduct a qualitative analysis to evaluate the contribution of each component in the Genre-Aware Dance Generation stage. As illustrated in Fig. 5, the Specialized Experts (SE), Universal Experts (UE), and the Mamba-enhanced backbone (Mamba) each play a crucial role in shaping the quality of the generated dance motions.

Removing SE, which is responsible for capturing genre-specific stylistic features, results in a significant loss of genre identity. This removal leads to mismatches where, for example, soft or fluid movements are applied to intense, percussive music, breaking the stylistic coherence expected for the genre. In contrast, the exclusion of UE impacts the overall complexity of the generated motion. Without UE, the generated sequences tend to be overly simplistic, often consisting of static poses or repetitive, monotonous movements, such as constant hand-raising, which lack the dynamic variation essential for engaging dance sequences. Furthermore, omitting the Mamba module, which enhances the backbone with a selection mechanism and scan module, results in a significant decrease in both movement diversity and alignment with the music. The generated dances become less responsive to the rhythmic and dynamic changes in the music, leading to sequences that feel disjointed or fail to reflect the musical structure accurately.

Collectively, these observations highlight the importance of each component in the overall framework. The combination of SE, UE, and Mamba ensures that the generated dance is not only genre-appropriate but also rich in motion variety and tightly aligned with the music.

### A.3 Computational Analysis

Table 5: Comparison for Computational Analysis.

| Model | Parameters | Run Time |
|---|---|---|
| Bailando++ | 152M | 5.46s |
| **FineNet** | **94M** | **3.97s** |
| Lodge | 235M | 4.57s |
| MEGADance | 120M | 4.31s |

In this section, we present the computational analysis of our approach. Providing quantitative comparisons of parameters and runtime helps clarify the efficiency advantages of our design. We report results for generating a 1024-frame (34.13s) dance sequence. Notably, MEGADance employs sparse expert activation, where only one specialized expert is activated per input, substantially reducing the effective computation during inference. "Parameters" are only calculated the activated parts. All "Run Times" are conducted on an RTX 3090 GPU with an Intel Xeon Gold 5218 CPU. As shown in Table 5, MEGADance exhibits slightly higher complexity than FineNet but delivers significantly better generation quality (see Tables 13). Compared to Bailando++ and Lodge, it achieves lower latency and requires fewer parameters while maintaining superior performance. These results highlight MEGADances favorable balance between efficiency and quality.

### A.4 Scalability Analysis

Table 6: Scalability Analysis of MEGADance.

| Model | $FID_k \downarrow$ | $FID_g \downarrow$ | $FID_s \downarrow$ | $DIV_k \uparrow$ | $DIV_g \uparrow$ | $DIV_s \uparrow$ | BAS↑ |
|---|---|---|---|---|---|---|---|
| MEGADance | 50.00 | 13.02 | 2.52 | 6.23 | 6.27 | 5.78 | 0.226 |
| MEGADance (-Dunhuang) | 54.70 | 14.10 | 2.41 | 6.35 | 6.12 | 5.61 | 0.224 |
| MEGADance (-Breaking) | 51.90 | 13.30 | 2.60 | 6.18 | 6.24 | 5.94 | 0.211 |

In this section, we discuss the scalability of MEGADance. (1) The FSQsin the HFDQ stage operates without retraining when new genres or samples are introduced. The motion patterns in the FineDance dataset are sufficiently diverse and representative to support generalization. (2) As described in Section 3.3.1, MEGADance employs a Mixture-of-Experts (MoE) design in which Specialized Experts are selected according to discrete genre labels. This modular architecture inherently supports genre extension. Specifically: (i) the General Expert and existing Specialized Experts can be preserved; (ii) a new Specialized Expert can be added and trained exclusively on a subset containing the new genre; and (iii) only the genre-to-expert mapping in the routing policy needs to be updated, without requiring structural modifications. (3) To verify this capability, we performed an additional experiment in which the Dunhuang and Breaking genres are removed from the original 16-genre training set, and subsequently reintroduced following the above extension strategy. As

shown in Tab. 6, the performance degradation relative to full retraining is negligible, indicating that MEGADance can efficiently incorporate new genres without complete retraining.

## A.5 Mixed Genre Generation

Figure 6: Visualization.

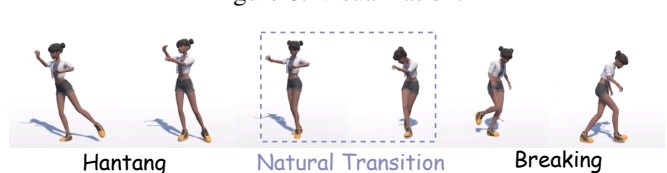

Hantang      Natural Transition      Breaking

Figure 7: User Study.

| Model | SQ↑ | MQ↑ |
|-------|------|------|
| FineNet | 2.81 | 4.12 |
| Lodge | 3.14 | **4.27** |
| MEGADance | **4.58** | 4.25 |

**Note:** *Analysis for dance generation under mixed genre condition.*

In this section, we evaluate the ability of MEGADance to generate dances under mixed-genre conditions. Thanks to its autoregressive generation manner, MEGADance can maintain motion continuity even when the accompanying music contains transitions between distinct genres. For example, in a challenging case transitioning from HanTang to Breaking, MEGADance not only produces highly coherent transitions but also preserves the distinctive stylistic characteristics of each genre before and after the switch, as shown in Fig. 6. To further assess this capability (an essential aspect of genre controllability) we conducted an additional user study using the same music tracks and participant setup as described in Section 4.2. In each test sequence, the first half is conditioned on *genre 1* and the second half on *genre 2*, with genre pairs randomly sampled. Participants rated the generated dances on a 5-point scale for *Style Quality* (SQ: alignment with the target genre) and *Motion Quality* (MQ: smoothness and naturalness). As summarized in Tab. 7, MEGADance achieves high MQ scores and clearly outperforms other methods in SQ for mixed-genre generation.

## A.6 Questionnaire for User Study

User feedback is essential for evaluating generated dance movements in the music-to-dance generation task, due to the inherent subjectivity of dance [45]. We select 30 real-world music segments, each lasting 34 seconds, and generated dance sequences using the models described above. These sequences are evaluated through a double-blind questionnaire completed by 30 participants with dance backgrounds, including undergraduate and graduate students. Participants are compensated at a rate exceeding the local average hourly wage. The questionnaires used a 5-point scale (Great, Good, Fair, Bad, Terrible) to assess three aspects: Dance Synchronization (DS, alignment with rhythm and style), Dance Quality (DQ, biomechanical plausibility and aesthetics), and Dance Creativity (DC, originality and range). The screenshot of our user study website is shown in Fig. 8, displaying the template layout presented to the participants. In addition to the main trials, participants are also subjected to several catch trials, which involved displaying Ground Truth videos and videos with distorted motion. Participants who failed to rate the GT videos higher and the distorted motion videos lower are considered unresponsive or inattentive, and their data are excluded from the final evaluation.

## A.7 Future Work

**Customized Dance Generation** While our current work successfully enables genre-aware control in music-to-dance generation, genre labels inherently impose rigid constraints and offer limited flexibility for user intent expression. Existing controllable generation approaches remain insufficiently expressive for practical deployment [31, 11, 7]. In future work, we plan to extend control modalities beyond predefined genre categories by incorporating free-form textual descriptions. Compared to genre labels, text allows users to articulate choreography requirements in a more intuitive and nuanced manner, facilitating personalized and expressive dance generation. This direction not only enhances user interactivity and creativity but also opens up new opportunities for content-driven applications in virtual performance and human-computer interaction.

**Noise-Resistant Dance Generation** 3D motion capture data often suffer from noise artifacts such as sudden positional jumps or temporal discontinuities, as observed even in high-quality datasets like

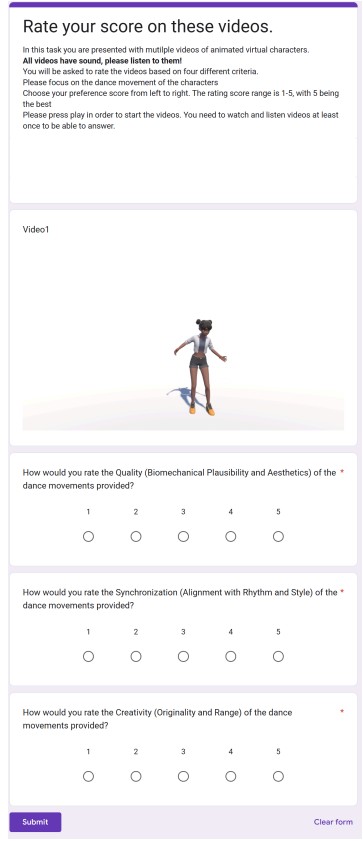

Figure 8: The screenshots of user study website for participants.

FineDance[2]. Moreover, the limited scale of 3D dance datasets makes models prone to overfitting. Future research should explore robust architectures and data augmentation strategies that maintain motion plausibility and stylistic coherence under noisy or incomplete input, thereby improving the reliability and generalization of music-to-dance generation systems.

