# OpenReview forum: "MEGADance: Mixture-of-Experts Architecture for Genre-Aware 3D Dance Generation"
_NeurIPS.cc/2025/Conference — NeurIPS 2025 poster_

### Official Review · Reviewer_Yz76 · 2025-06-02

**Clarity:** 3
**Significance:** 3
**Originality:** 3
**Rating:** 5
**Confidence:** 3

**Summary:**

In this work, the authors introduce MEGADance, a novel two-stage architecture for music-driven 3D dance generation that emphasizes genre-aware choreography. This is achieved by decoupling dance generality and genre specificity across two stages. In the first stage, MEGADance uses Finite Scalar Quantization to learn a diverse motion codebook. Combined with kinematic and dynamic constraints, FSQ improves reconstruction fidelity and motion plausibility. In the second stage, the MEGADance uses a Mixture-of-Experts (MoE) model, where Universal Experts model genre-invariant motion patterns and Specialized Experts model genre-specific patterns, via hard routing genre labels. Each expert is implemented as an autoregressive Mamba-Transformer, enabling both local and global temporal modeling. In part because of its strong inductive biases and genre sensitivity, MEGADance outperforms prior methods across quantitative metrics as well as human evaluations.

**Questions:**

1. Could authors provide any information on model parameter complexity? The stated generation runtime of 0.19 second is helpful but it would be better to understand the computational cost of the model as a whole, especially as a comparison to previous models. More broadly, quoting the NeurIPS Paper Checklist (line 607), authors should discuss the computational efficiency of the proposed algorithms and how they scale with dataset size.

2. How well does MEGADance scale when new genres are added?
Since the Specialized Experts are tied to specific genres, it's unclear how the model handles genre extension without full retraining. Could experts be modularly added or fine-tuned?

3. Can MEGADance generate dances that blend multiple genres?
In its current form, hard routing limits the model to one genre at a time. Authors mention in App4 that free form text would be an interesting next step to allow for more creative generation... but I am curious to hear if there could be an even simpler next step. For example, would it be possible to mix Specialized Experts' latents? This reminds me of the field of style transfer, that studies this exact sort of exploration of the latent space, ex. Yin et. al https://arxiv.org/abs/2208.09406.

**Ethical Concerns:**

["NO or VERY MINOR ethics concerns only"]

**Final Justification:**

The authors address my (generally lightweight) concerns in a satisfactory manner. However, I don't believe the paper is ``groundbreaking" in such a way that would warrant raising my score to a 6.

**Limitations:**

A limitations section is included in App4, although I would like to see it padded with information that I have asked about above, such as computational cost, scaling with new data, and more flexible generation modes.

**Quality:**

3

**Strengths And Weaknesses:**

**Quality**
- S: Overall this is a high-quality paper. It is well-written, proposes novel and strong methodology, and performs extensive experiments with convincing quantitative and qualitative results.
- W: See question 1 about complexity analysis; this is important information that appears to be totally left out. Also, the experiments don't show any error bars, although this appears to be the case for other papers in the field. Could the authors expand on this?
- W: I would have appreciated if the authors could share the code for the purposes of evaluating how reproducible it seems. I understand anonymity concerns prevent a GitHub link, but there are other ways of sharing code anonymously, such as using a code anonymizing website like https://anonymous.4open.science/.

**Significance / Originality**
- S: The model carries real-world relevance from a creative standpoint, as it enables genre-aware music-generated choreography, albeit only for genres cleanly labelled in training data.
- S: MoE + FSQ + Mamba-Transformer is novel in this domain.
- W: It remains unclear how well MEGADance generalizes or adapts to new genres (i.e., how simple would this be? how expensive would this be?). This imposes limits on the architecture and its use cases in any applied scenario. See Q2 / Q3 on this point as well.

**Clarity**
- S: The text is well written overall and the math notation is consistent.
- W: This is minor, but I would appreciate better figure captions. For example, the master architecture figure (Fig. 2) should have a clear caption that walks the reader through Stage 1, then Stage 2, and breaks down the components of the figure. It is only clear to me where the blue and pink boxes fit in with respect to the left box after I have studied the methodology, whereas it should be pretty obvious to someone just skimming the paper.

---

> ### Author Rebuttal · Authors · 2025-07-30
>
> **To Reviewer Yz76:**
>
> **Q1: Suggestions on providing the complexity and scalability analysis.**
> **A1:** Thank you for the insightful question. Beyond generation latency, we agree that analyzing model complexity and scalability is essential, and we make following clarifications:
> 1. In practical deployment scenarios, runtime latency is typically the most critical metric, which is why we reported it in Line 279 (Section 4.2). MEGADance achieves 0.19s per second of generated dance, enabling real-time applications.
> 2. As you pointed out, complexity analysis is also important. Thus, we include a comparison of complexity for generating a 1024-frame (34.13s) dance sequence. Note that MEGDance activates only the relevant experts rather than all experts when computing "Parameters". All "Run Time" evaluations were conducted on an RTX 3090 GPU with an Intel Xeon Gold 5218 CPU. As summarized below, MEGDance has slightly higher complexity than FineNet but substantially better generation quality (refer to Tables 1-3 for details). Compared to Bailando++ and Lodge, it has lower latency and fewer parameters while maintaining superior performance. These results demonstrate MEGDance's favorable trade-off between efficiency and quality. We will include this analysis in the revision.
> | Model | Parameters↓ | Run Time ↓ |
> |-------|-------------|------------|
> | Bailando++ | 152M | 5.46 s |
> | FineNet | 94M | 3.97 s |
> | Lodge | 235M | 4.57 s |
> | MEGDance | 120M | 4.31 s |
> 3. Regarding scalability with dataset size, MEGADance exhibits favorable scaling properties. Its two-stage structure benefits from increased training data: the FSQ-based quantization achieves better codebook coverage, while the genre-aware MoE design enables each expert to specialize more effectively. Moreover, the sparse routing in our MoE ensures that only one expert is activated per input, keeping computational cost nearly constant regardless of the total number of genres. We have evaluated performance of MEGADance on two large-scale datasets (FineDance and AIST++), and observed stable and generalizable performance across varied genre distributions, as shown in Tables 1 and 2.
>
> **Q2: Suggestion on sharing the code.**
> **A2:** Thank you for the helpful suggestion. We will release the MEGDance code to benefit the community. As NeurIPS policy prohibits external links in the rebuttal, the code will be made publicly available in the final revision.
>
> **Q3: Concerns about the generalization ability of MEGADance to unseen genres**
> **A3:** Thank you for the insightful question. As discussed in Section 3.3.1 (Lines 176-192), MEGADance adopts a Mixture-of-Experts (MoE) design where Specialized Experts are selected based on discrete genre labels. This modular architecture inherently supports genre scalability. Specifically: (1) The General Expert and existing Specialized Experts can be retained; (2) A new Specialized Expert can be added and trained solely on a subset containing the new genre; (3) Only the genre-to-expert mapping in the routing policy needs to be updated—no structural changes are required.
> To validate this capability, we conducted an additional experiment: we removed Dunhuang and Breaking from the original 16-genre training set, and then reintroduced each genre using the extension strategy stated above. As shown below, the performance gap compared to full retraining is negligible, demonstrating MEGDance's ability to handle genre extension effectively without requiring full retraining. We will include this result in the revision.
> | Model | FID_k↓ | FID_g↓ | FID_s↓ | DIV_k↑ | DIV_g↑ | DIV_s↑ | BAS↑ |
> |-------|--------|--------|--------|--------|--------|--------|------|
> | MEGADance | 50.00 | 13.02 | 2.52 | 6.23 | 6.27 | 5.78 | 0.226 |
> | MEGADance (-Dunhuang) | 54.70 | 14.10 | 2.41 | 6.35 | 6.12 | 5.61 | 0.224 |
> | MEGADance (-Breaking) | 51.90 | 13.30 | 2.60 | 6.18 | 6.24 | 5.94 | 0.211 |
>
> **Q4: Question on MEGADance's ability to generate dances that blend multiple genres.**
> **A4:** Thank you for the insightful question. Here is our answer:
> 1. As you suggested, all experts in MEGDance share a unified codebook, so the latent tokens (i.e., FSQ indices) produced by different Specialized Experts reside in the same space. This allows direct addition, interpolation, and fusion of latent representations, making mixed-genre dance generation a viable and promising direction.
> 2. Furthermore, MEGDance’s autoregressive generation scheme ensures strong motion continuity, even when switching genres mid-sequence (e.g., from Hangtang to Breaking). As NeurIPS policy prohibits external links in the rebuttal, we will include corresponding visualizations in the revision.
> 3. To evaluate this capability—an essential aspect of genre controllability—we conducted an additional user study using the same music and participant setup as shown in Section 4.2. Specifically, the first half of each sequence was conditioned on genre1 and the second half on genre 2, where genre pairs were randomly sampled. Participants rated the results on a 5-point scale for Style Quality (SQ: alignment with target genre) and Motion Quality (MQ: smoothness and naturalness). As summarized below, MEGDance achieves high MQ and clearly outperforms other methods in SQ for mixed-genre generation. We will include this in the revision.
> | Model | SQ | MQ |
> |-------|----|----|
> | FineNet | 2.81 | 4.12 |
> | Lodge | 3.14 | 4.27 |
> | MEGADance | 4.58 | 4.25 |
>
> **Q5: Concerns about the absence of variance reporting in experiments.**
> **A5:** Thank you for the insightful question. Here is our answer: (1) In music-to-dance generation, prior works [1,2] typically report mean performance without variance metrics, as evaluations are based on deterministic inference under fixed settings. We followed this convention to ensure comparability. (2) However, we agree that reporting variance is valuable. We now additionally provide standard deviations of our main evaluation metrics below. We will include it in the revision.
> | Model | FID_k ↓ | FID_g ↓ | FID_s ↓ | DIV_k ↑ | DIV_g ↑ | DIV_s ↑ | BAS ↑ |
> |-------|----------|----------|----------|----------|----------|----------|--------|
> | MEGADance | 50.00 ± 4.42 | 13.02 ± 3.67 | 2.52 ± 0.43 | 6.23 ± 0.21 | 6.27 ± 0.10 | 5.78 ± 0.09 | 0.226 ± 0.005 |
>
> [1] Li, Ronghui, et al. "Lodge: A coarse to fine diffusion network for long dance generation guided by the characteristic dance primitives." Proceedings of the IEEE/CVF Conference on Computer Vision and Pattern Recognition. 2024.
> [2] Li, Ronghui, et al. "Finedance: A fine-grained choreography dataset for 3d full body dance generation." Proceedings of the IEEE/CVF International Conference on Computer Vision. 2023.
>
> **Q6: Suggestions the clarity and readability of Fig. 2.**
> **A6:** Thank you for the helpful suggestion. We will revise the caption of Fig. 2 to improve clarity and readability, as follows:
> Figure 2: Overview of MEGADance. Stage 1 (HFDQ) quantizes dance into upper/lower-body latent codes using FSQ with kinematic and dynamic reconstruction constraints. Stage 2 (GADG) maps music to codes via an $L$-layer Mixture-of-Experts architecture (blue: specialized, pink: universal), where each expert is equipped with a Mamba-Transformer hybrid backbone.

---

> > ### Comment · Reviewer_Yz76 · 2025-08-01
> >
> > I thank the authors for their comprehensive responses.
> >
> > Q1 :
> > - Complexity: I find the reported model complexity convincing.
> > - Scalability: I am encouraged by the authors' explanations on how the architecture scales with dataset size. However, I would appreciate if the authors could quantify this scaling, rather than using qualitative terms like "nearly constant." What is the added computational cost per added genre and/or added data sample? I think making this information readily available in the paper would greatly improve its attractiveness to practitioners.
> >
> > Q3: I am encouraged by the newly reported results on generalizing to unseen genres. I am also encouraged by the seemingly low cost of the extension strategy -- perhaps this would be a good way to quantitatively measure scaling (in reference to Q1)?
> >
> > Q4: I find the newly reported results on multi-genre sampled sequences convincing. I look forward to seeing the visualizations in the released version.
> >
> > Q5: Thank you for reporting standard deviations, even if not common in the field.

---

> > > ### Author Response · Authors · 2025-08-06
> > > **Scalability Clarification for MEGADance**
> > >
> > > Thank you for the encouraging feedback. We address your interest in the scalability aspects of MEGADance as follows:
> > >
> > > (1) The FSQ module in the HFDQ stage does not require retraining when new genres or samples are introduced. The motion patterns covered by the FineDance dataset are already diverse and representative enough to ensure generalizability.
> > >
> > > (2) During inference, adding new genres or samples does not affect runtime efficiency, as the SE module activates only one expert at a time via hard routing.
> > >
> > > (3) Following the protocol described in Q3, we freeze the parameters of the Universal Expert (UE) and existing Specialized Experts (SE), and fine-tune a new expert on the target genre for 5 epochs. For example, incorporating the Breaking genre (14.60 minutes of data) required only 5.3 minutes of fine-tuning. Similarly, adding the Dunhuang genre (4.26 minutes of data) took just 2.3 minutes. These results demonstrate the modular extensibility and low computational overhead of our architecture.
> > >
> > > We will include this discussion in the final revision.

---

### Official Review · Reviewer_1pZx · 2025-07-02

**Clarity:** 2
**Significance:** 3
**Originality:** 3
**Rating:** 4
**Confidence:** 4

**Summary:**

The authors propose a two-stage music-to-dance generation framework that integrates a Mixture-of-Experts (MoE) architecture, Finite Scalar Quantization (FSQ) with kinematic-dynamic dual constraints, and a Mamba-Transformer hybrid backbone. The model is designed to generate high-fidelity and genre-controllable dance motions.

**Questions:**

1.	Many prior works have pointed out that autoregressive frameworks often suffer from error accumulation and motion freezing [8], how did the authors address this issue?

2.	Please refer to the weaknesses section.

**Ethical Concerns:**

["NO or VERY MINOR ethics concerns only"]

**Final Justification:**

The authors' rebuttal has addressed most of my concerns. After reading the other reviewers' comments, I believe the manuscript is generally suitable for publishing in NeurIPS. However, the authors may need to address all the comments carefully in the camera-ready version. In this case, I remain with my original recommendation.

**Limitations:**

Yes

**Quality:**

3

**Strengths And Weaknesses:**

The paper introduces a two-stage music-to-dance generation model that combines MoE, FSQ, and a Mamba-Transformer backbone to improve motion fidelity and genre controllability.

Strengths

(1)	By using a Mixture-of-Experts (MoE) to separate genre-specific and shared motion patterns, and combining it with Finite Scalar Quantization (FSQ) and a Mamba-Transformer hybrid backbone, the model achieves better choreographic consistency and motion fidelity. This combination is both original and novel, offering an effective solution for controllable and high-fidelity dance generation.

(2)	The paper presents solid experimental comparisons and ablation studies, supported by comprehensive visualizations to validate the effectiveness of each component.

Weaknesses:

(1)	While the authors claim to compare against multiple methods, Table 1 only includes two recent baselines: FineDance (2023) and LODGE (2024), with no broader comparison to other relevant or competitive approaches. Moreover, it is unclear why Table 2 omits standard metrics FID_s and DIV_k for the AIST++ evaluation.

(2)	Although the paper claims that both the MoE design and the Mamba-Transformer backbone improve computational efficiency, only the inference time of the proposed model is reported. It would strengthen the argument to provide FLOPs or latency comparisons with other models, or at least include ablation results showing the impact of these modules on efficiency.

(3)	The loss formulation involving velocity and acceleration (Eq. 3) lacks clarity. Notations such as S’ and S’’ are introduced without explanation. It would be helpful to rewrite the equation in a more explicit form.

---

> ### Author Rebuttal · Authors · 2025-07-30
>
> **To Reviewer 1pZx**
>
> **Q1: Concerns on the evaluations on AIST++.**
> **A1:** Thank you for the helpful suggestion. We clarify as follows:
> 1. *Table 2* contains a typo and it actually represents $DIV_k$ and $DIV_g$.
> 2. *FineDance*, as the largest dataset with more diverse genres, already provides strong evidence of *MEGADance*'s superiority through its competitive $FID_s$ and $DIV_s$ results. And most baseline methods on *AIST++* do not incorporate genre conditioning (except *Lodge*), making direct comparisons on $FID_s$ and $DIV_s$ potentially unfair.
> 4. Following your suggestion, we additionally report *AIST++* results on $FID_s$ and $DIV_s$. As summarized below, *MEGADance* substantially outperforms all baselines. We will include this in the revision.
>
> | Model       | FID_s | DIV_s |
> |-------------|-------|-------|
> | GT          | 0.00  | 7.24  |
> | FACT        | 33.80 | 4.68  |
> | Bailando++  | 22.75 | 4.32  |
> | EDGE        | 19.63 | 5.37  |
> | Lodge       | 13.85 | 5.93  |
> | MEGADance | **7.12** | **6.76** |
>
> **Q2: Suggestions on providing the computational cost comparison or ablation.**
> **A2:** Thank you for the helpful suggestion. We make clarification as follows:
> 1. In practical applications, users usually prioritize *inference latency* (reported in Line 279, Section 4.2). *MEGADance* achieves **0.19 seconds per second** of generated dance, enabling real-time responsiveness.
> 2. We agree that providing additional computational cost analysis, such as Parameters and Run Time comparisons, helps clarify the efficiency benefits of our design. We now include a comparison for generating a 1024-frame (34.13s) dance sequence. Importantly, due to sparse expert activation, MEGADance only activates one Specialized Expert per input, which significantly reduces the effective computation during inference. All “Run Time” evaluations were conducted on an RTX 3090 GPU with an Intel Xeon Gold 5218 CPU. As summarized below, MEGDance has slightly higher complexity than FineNet but substantially better generation quality (refer to Tables 1-3). Compared to Bailando++ and Lodge, it achieves lower latency and fewer parameters while maintaining superior performance. These results reflect MEGDance's favorable trade-off between efficiency and quality. We will include this analysis in the revision.
>
> | Model       | Parameters(↓) | Run Time(↓) |
> |-------------|---------------|-------------|
> | Bailando++  | 152M          | 5.46s      |
> | FineNet     | **94M**           | **3.97s**      |
> | Lodge       | 235M          | 4.57s      |
> | MEGADance | *120M*     | *4.31s*  |
>
> 3. Since MEGADance already achieves real-time performance, our ablation study primarily focuses on generation quality and controllability rather than parameters or runtime, as these aspects are more critical for evaluating the effectiveness of our design—particularly in genre-aware choreography generation. However, we agree that computational efficiency is an important perspective and will discuss it in the revision.
>
> **Q3: Suggestions on the statement of Eq. 3.**
> **A3:** Thank you for the helpful suggestion. In *Eq. 3*:
> - `S′` and `S″` denote first- and second-order derivatives of `S`
> - `J′` and `J″` denote first- and second-order derivatives of `J`
> We will clarify these in the revision.
>
> **Q4: Question about the exposure bias in autoregressive manner.**
> **A4:** Thank you for the insightful question. Here is our response:
> 1.In practice, exposure bias does not always have a significant negative impact in motion generation tasks, especially when strong temporal models (e.g., Transformer or Mamba) and well-structured conditions are employed—models trained with cross-entropy loss can still produce high-quality sequences.
> 2. Following your suggestion, we implemented Scheduled Sampling (SS) to mitigate exposure bias. As shown in the table below, some metrics even slightly degrade with SS. We will include this discussion in the revision.
>
> | Model          | FID_k↓ | FID_g↓ | FID_s↓ | DIV_k↑ | DIV_g↑ | DIV_s↑ | BAS↑ |
> |----------------|--------|--------|--------|--------|--------|--------|------|
> | MEGADance      | 50.00  | 13.02  | 2.52   | 6.23   | 6.27   | 5.78   | 0.226|
> | MEGADance (SS) | 42.36  | 16.21  | 3.47   | 5.18   | 7.04   | 6.33   | 0.228|

---

> > ### Comment · Reviewer_1pZx · 2025-08-05
> >
> > The authors have addressed most of my comments in a detailed response. Thanks.

---

### Official Review · Reviewer_xz7a · 2025-07-02

**Clarity:** 3
**Significance:** 3
**Originality:** 3
**Rating:** 5
**Confidence:** 4

**Summary:**

This paper presents a system for music-conditioned and genre-aware dance motion generation. It contains some key innovative components. Finite scalar quantization (FSQ) is used to encode motions into latent space representations. Mixtures of genre-specialized and universal experts are then used to generate genre-aware dance motion sequences. To capture both long-range dependencies and finer local variations in motion, these experts are implemented using combinations of Mamba and Transformers. Through a number of experiments, the proposed system was evaluated with key characteristics highlighted.

**Questions:**

The paper includes some rather specific qualities of what the specialized and universal experts are expected to capture, but how can you know that these are true? The only difference seems to be a switch based on the genre when selecting the specialized expert.

Where do the numbers for Bailando++ come from, given that they seem to be different from the original paper or what has been published in other papers? Is it related to the fact that this method is not genre-aware?

**Ethical Concerns:**

["NO or VERY MINOR ethics concerns only"]

**Final Justification:**

The rebuttal addressed the few concerns raised in good a manner. There is thus no reason to change recommendation.

**Limitations:**

yes

**Paper Formatting Concerns:**

There seems to be no formatting concerns.

**Quality:**

4

**Strengths And Weaknesses:**

The proposed system is shown to be very competitive compared to the state-of-the-art when tested on the two most common datasets. The best results are shown in the genre controllability evaluation, which is indeed what the paper aims to improve. A rather extensive ablation study further highlights the importance of the different components of the system.

The paper is very well written and structured, which includes illustrations. Even if it contains a lot of content, it is still rather easy to read and understand. The proposed architectural components are both easy to understand and make perfect sense. One of the more intriguing components of the proposed system is FSQ for quantization. FSQ is more lightweight than VQ-VAE and doesn't require a codebook to be trained, a codebook that may be poorly utilized. In the ablation study provided in the paper, FSQ is shown to perform better than VQ-VAE, despite its simplicity.

Another characteristic component that makes sense, and is supported by positive results, is the distinction between specialized and universal experts. Even if there are many different genres in dance, there are few enough to motivate a specialized expert for each, rather than having a single generic generator and the genre as an argument that conditions the output.

The combination of Mamba and Transformers is not novel, but appears to be suitable for the task. Transformers are known to be good at handling long-range dependencies and interaction between music and motion, while Mamba may be better at capturing local variations in motion. The benefit of the combination is clear from the reported results.

In the comparison between different methods, the number of tested methods is quite low. There are more methods in the literature than those shown in the paper, especially if you look back in time. Those included, however, are representative of the current state-of-the-art. The numbers seem to have come from the original papers, except for Bailando++, where the numbers are worse than what has been reported earlier. There ought to be a good reason for this to be clarified.

Minor things: The captions of Fig. 4a and Fig. 4b seem to be switched. Sometimes there are spaces before citations, and sometimes not. It would be better to make it more uniform. There is a missing bold value for FACT in Table 2. In the same table, it ought to be $DIV_k$ and $DIV_g$, not $DIV_g$ and $DIV_s$.

---

> ### Author Rebuttal · Authors · 2025-07-30
>
> **To Reviewer xz7a:**
>
> **Q1: Questions about the experimental result of Bailando++.**
> **A1:** Thank you for the insightful question. Here is our answer:
> 1. The reported results for Bailando++ are obtained using the publicly available official code, which we retrained and evaluated following the instructions. Despite our efforts to replicate the original experimental setup, we were unable to reproduce the exact numbers reported in the paper. Therefore, we report our evaluation results for fair and consistent comparison across all baselines under the same environment. We will explicitly annotate in the revised table that these numbers are from our reproduction.
> 2. Regarding your question about genre awareness: it is true that Bailando++ does not incorporate genre conditioning, which may partially explain its lower performance compared to genre-aware methods like MEGADance. However, this difference in model design does not account for the discrepancy between our reproduced results and those reported in the original paper. We will clarify it in the revision.
>
> **Q2: Typos in the paper.**
> **A2:** Thank you for pointing them out. The following is our response:
> 1. We acknowledge the typo in *Table 2*—column headers should be $DIV_k$ and $DIV_g$.
> 2. The $DIV_g$ value of FACT in *Table 2* should indeed be bolded.
> 3. We will standardize citation spacing throughout the paper.
> 4. We would like to point out that *Figure 4* does not contain subfigures (a) and (b).
>
> These typos will be corrected in the revision.
>
> **Q3: Concerns on whether specialized and universal experts truly capture distinct roles as intended.**
> **A3:** Thank you for the insightful question. Here is our answer:
> 1. $FID_k$ and $FID_g$ primarily reflect universal dance quality, while $FID_s$ captures genre-specific fidelity. As shown in *Table 4*, removing the Universal Expert (*UE*) notably degrades $FID_k$ and $FID_g$, whereas removing the Specialized Expert (*SE*) leads to a sharp drop in $FID_s$—indicating their distinct contributions.
> 2. We further conducted a *t-SNE* analysis on the final-layer features of *SE* and *UE*. Visualizing genre-specific color-coded clusters, *SE* features form well-separated clusters while *UE* features do not—empirically supporting their intended functional distinction. As figures are not allowed during the rebuttal phase, we only provide this textual description.
>
> We will include both figure and explanation in the revision.

---

> > ### Comment · Reviewer_xz7a · 2025-08-08
> >
> > Thanks for the information provided in the rebuttal, especially related to the respective roles of experts. The comment regarding the subfigures of Figure 4 in the review was incorrect, since it relates to Table 4, not Figure 4.

---

> > > ### Author Response · Authors · 2025-08-08
> > > **Writing typo in Tab. 4**
> > >
> > > Thanks. You are correct, and we will fix it in the revision.

---

### Official Review · Reviewer_oRpN · 2025-07-03

**Clarity:** 3
**Significance:** 2
**Originality:** 2
**Rating:** 4
**Confidence:** 3

**Summary:**

This paper proposes an MOE-based architecture MEGADance for 3D music-to-dance generation by decoupling it into the dance content part and the genre style part. MEGADance consists of two stages - the High-Fidelity Dance Quantization Stage (HFDQ) and the Genre-Aware Dance Generation Stage (GADG). HFDQ learns the motion representation as discrete tokens through 3D motion reconstruction. Then the GADG generates motion tokens based on music features, genre features, and previously generated motion tokens in a generative way. To enhance the autoregressive generation process, MoE and Mamba-Transformer are applied. Experiments on two benchmark datasets show MEGADance outperforms current baselines, especially in genre controllability.

**Questions:**

Please see weakness.

**Ethical Concerns:**

["NO or VERY MINOR ethics concerns only"]

**Final Justification:**

After reading the rebuttal, my concern on the motivation of the proposed method and complexity analysis has been solved.

**Limitations:**

Yes.

**Paper Formatting Concerns:**

No.

**Quality:**

3

**Strengths And Weaknesses:**

Strength:
- Experiments show that the proposed framework outperforms baselines on two datasets (FineDance and AIST++).
- The MoE design to decouple genre and dance content sounds plausible.

Weakness:
- It's a little bit difficult to judge the novelty of the proposed framework. MEGADance is based on Transformer autoregressive decoder backbone with the generation target being discrete motion tokens, which is a relatively common practice for vision generation task using autoregressive Transformer. Although MEGADance proposes to include MoE and Mamba-Transformer for the autoregressive backbone, the core idea still remains the same and the proposed module/strategy themselves are not novel either. Motivation and analysis of the necessity of the proposed strategies are needed to help better understand the contribution.
- The complexity analysis in time and memory of the proposed framework is missing.

---

> ### Author Rebuttal · Authors · 2025-07-30
>
> **To Reviewer oRpN:**
>
> **Q1: Concerns on the novelty of MEGADance.**
> **A1:** Thank you for the insightful comment. While autoregressive Transformer-based token generation is indeed a common paradigm, MEGADance introduces several key architectural distinctions beyond simple module application:
> 1.  **Structural Decoupling of Semantics**. Unlike monolithic sequence modeling in prior work, we explicitly decouple choreographic semantics into: Genre-invariant rhythmic structure (modeled by **Universal Expert**) and Genre-specific stylistic priors (modeled by **Specialized Experts**). This reflects real-world choreographic composition and improves both genre control and generalization.  Ablation (Table 4a): Removing either of these two structures significantly degrades genre fidelity or structural coherence.
> 2.  **Task-Specific Modular Routing via MoE**. MEGADance uses a **genre-conditioned hard routing policy** where each sample activates a single genre-specific expert, which differs fundamentally from Conventional sparse MoE (e.g., for language modeling) where activation is learned. Our approach encodes task-specific inductive bias and prevents expert collapse.
> 3.  **Hybrid Modeling of Cross/Intra-modal Dynamics**. We replace homogeneous Transformers with a **Mamba–Transformer hybrid**: Mamba captures fine-grained *intra-modal continuity* (especially motion), and **Transformer** handles long-range *cross-modal alignment* (music and dance). This addresses Transformers' lack of inductive bias for temporal locality. Ablation (Table 4a): Hybrid modeling design improves generation quality and synchronization.
> We will further clarify these points in the revision.
>
>
>
> **Q2: Suggestions on the complexity analysis.**
> **A2:** Thank you for the helpful suggestion. We make following clarification regarding this issue:
> 1.  **Runtime Latency Focus** . In practical applications, user priority is run-time latency (reported in Line 279, Section 4.2). This remains our primary efficiency metric.
> 2.  **Complexity Analysis Expansion**. As suggested, we now include comparison for generating **a 1024-frame (34.13s) sequence**. MEGADance activates only relevant experts (not all) during computation. All benchmarks run on **RTX 3090 GPU + Intel Xeon Gold 5218 CPU**. As summarized below, MEGDance has slightly higher complexity than FineNet but substantially better generation quality (refer to Tables 1-3). Compared to Bailando++ and Lodge, it has lower latency and fewer parameters while maintaining superior performance. These results reflect MEGDance's favorable trade-off between efficiency and quality. We will include this analysis in the revision.
>
> | Model        | Parameters (↓) | Run Time (↓) |
> |--------------|----------------|--------------|
> | Bailando++   | 152M           | 5.46s       |
> | FineNet      | **94M**            | **3.97s**       |
> | Lodge        | 235M           | 4.57s       |
> | *MEGADance* | *120M*       |  *4.31s*  |

---

> > ### Comment · Reviewer_oRpN · 2025-08-07
> >
> > Thanks for the rebuttal. I've carefully read the rebuttal and comments from other reviewers. I now decide to increase my rating toward acceptance.

---

> ### Comment · Area_Chair_6CfC · 2025-08-05
>
> Dear reviewer,
>
> Please read the rebuttal and start discussion with authors.
>
> AC

---

### Decision · Program_Chairs · 2025-09-17

**Decision:**

Accept (poster)

**Comment:**

This manuscript introduces MEGADance, a novel two-stage MoE architecture for genre-aware 3D dance generation, explicitly decoupling dance generality (universal expert) and genre specificity (specialized experts) via FSQ quantization and a Mamba-Transformer hybrid backbone. It achieves SOTA results and modular extensibility to new genres. Weaknesses initially cited (novelty, missing comparisons, scalability) were resolved in rebuttal. All reviewers converged on acceptance post-rebuttal, confirming the technical soundness and readiness of this work. Therefore, I recommend acceptance of this manuscript.